# An integrated multi-instrument methodology for studying marginal ice zone dynamics and wave-ice interactions

Sébastien Kuchly<sup>1</sup>, Baptiste Auvity<sup>1</sup>, Nicolas Mokus<sup>2</sup>, Matilde Bureau<sup>1</sup>, Paul Nicot<sup>3</sup>, Amaury Fourgeaud<sup>1</sup>, Véronique Dansereau<sup>2</sup>, Antonin Eddi<sup>1</sup>, Stéphane Perrard<sup>1</sup>, Dany Dumont<sup>3</sup>, and Ludovic Moreau<sup>2</sup>

**Correspondence:** Ludovic Moreau (ludovic.moreau@univ-grenoble-alpes.fr)

Abstract. Wave-driven fragmentation is a key mechanism shaping the Marginal Ice Zone (MIZ). Capturing this process is therefore essential for improving sea ice models, which currently do not fully capture the complex interactions between the forcing imposed by waves and the nonlinear dynamics of the resulting sea ice breakup and deformation. To investigate these interactions, we introduce a comprehensive multi-instrument dataset from a field campaign in the MIZ of the St .Lawrence Estuary, Canada, designed to characterize wave propagation and mechanical properties of sea ice under natural forcing conditions. The dataset integrates synchronized measurements from geophone arrays, wave buoys, smartphones configured as motion sensors, and Unmanned Aerial Vehicles (UAVs), all collected during coordinated deployments across diverse ice types and sea states.

Seismic data, recorded with geophone arrays, enable estimation of the ice thickness and elastic properties via active and passive wavefield analyzes. Concurrently, wave buoys and smartphones capture ocean wave characteristics including amplitude, wavelength, and attenuation near ice edges. UAV imagery is processed with advanced methods to detect vertical ice displacements with sub-centimetre sensitivity, allowing extraction of wave dispersion relations in different ice conditions. Preliminary analyzes demonstrate strong agreement between independent measurement methods, validating the dataset's quality. This multi-sensor approach offers unique opportunities to improve our understanding of wave-ice interactions, wave attenuation, and fracture dynamics in situ, thus offering a valuable resource for the sea ice and oceanographic research community to gain insight in wave-induced ice break-up mechanisms under natural conditions.

## 1 Introduction

Understanding the dynamics of sea ice in the changing climate represents a major challenge requiring multidisciplinary and international collaborations, especially in light of the prospect of a seasonally ice-free Arctic. Twenty years ago, this was projected to occur around 2100 (Parry et al., 2007), with a summer minimum sea ice extent declining at a rate of 10% per decade (Comiso et al., 2008). Recently, Kim et al. (2023) brought this timeline forward, projecting a seasonally ice-free Arctic between 2030 and 2050, regardless of greenhouse gas emission scenarios. This alarming acceleration alone is enough to justify

<sup>&</sup>lt;sup>1</sup>PMMH-ESPCI, Paris, France

<sup>&</sup>lt;sup>2</sup>Institut des Sciences de la Terre, Université Grenoble Alpes, Grenoble, France

<sup>&</sup>lt;sup>3</sup>Institut des sciences de la mer, Université du Québec à Rimouski, Québec G5L 3A1 Canada

the need to improve our understanding of sea ice dynamics. Compounding this prediction is the reality that the remaining ice in the Arctic is younger, thinner, more fragile and therefore more likely to be fragmented by ocean waves (Stroeve et al., 2012; Stroeve and Notz, 2018; Thomson, 2022; Horvat, 2022). This weakening feedback impacts the area where ocean swells or wind-generated waves can penetrate and therefore fragment the ice cover, known as the Marginal Ice Zone (MIZ). In the Arctic, the MIZ is projected to extend relative to the area covered by consolidated ice (Aksenov et al., 2017; Rolph et al., 2020; Song et al., 2025).

In the MIZ, waves break the ice into fragments, or floes, of smaller sizes. The associated evolution of the Floe Size Distribution (FSD) modifies the sea ice rheology (Dumont, 2022) – how the ice cover deforms under geophysical forcings – and triggers nonlinear feedback processes, for example by increasing melt rates and dynamical interactions with ocean currents and eddies (Horvat et al., 2016; Gupta et al., 2024). Besides that, spatial variations in ice and snow thickness modify the way light is filtered through sea ice, which influences the albedo, as well as phytoplankton bloom, both in the Arctic (Ardyna et al., 2014) and the Antarctic (Taylor et al., 2013). In this context, accurate monitoring of sea ice thickness and rheology is crucial for understanding the fragmentation of the ice cover by waves, in the view of forecasting its future state in polar to mid-latitude regions.

Large-scale monitoring of sea ice extent and concentration relies on low resolution ( $\mathcal{O} \sim 10^3 - 10^4$  m) satellite-borne microwave radiometers, while ice types, form, and thickness proxies can be derived or inferred at higher resolution from Synthetic Aperture Radar (SAR) imagery. Typical satellite-derived estimations of the ice thickness result essentially in an apparent, averaged value at the kilometre scale, because its estimation is sensitive to many local-scale factors, such as ice density, melt ponds, surface roughness, snow cover or concentration from CryoSat2 or IceSat2. In the absence of ground truth data, these lead to large uncertainties that are not compatible with mid-term predictions of the rapid evolution of sea ice. Therefore, obtaining accurate, multi-scale estimates of ice parameters will establish a new benchmark for sea ice monitoring, with impacts on many aspects of oceanographic and climate research (Kern et al., 2018).

Recently, observations of the interaction between waves and the marginal ice zone (MIZ) have been made possible — for example, in Svalbard using arrays of OpenMetBuoys (Rabault et al., 2024), or in Baffin Bay and the St. Lawrence Estuary (Canada) using unmanned aerial vehicles (UAVs) to video-record sea ice break-up caused by waves generated by the Canadian Coast Guard Ship (CCGS) *Amundsen* (Dumas-Lefebvre and Dumont, 2023a). The latter has proven to be a new and promising methodology for understanding wave-induced sea ice fracture. However, the quantitative assessment necessary to build, test, and develop predictive models and well-informed parametrization cannot be achieved without combining these aerial data with a characterization of the mechanical properties of the ice.

In this paper, we introduce a multi-instrument dataset that combines aerial video recordings with data measured on the ice with a variety of instruments: geophones, wave buoys and smartphones controlled remotely. With specific data processing for each instrument type, we show that it is possible to quantitatively assess ice breakup while simultaneously measuring its thickness and mechanical properties.

**Table 1.** Field experiments conducted during the 2024 BicWin campaign, with the corresponding ice type and sea state. See Figure 1 for site location. Instruments: D for drone, B for buoys, G for geophone and S for smartphones.

| Date        | Location         | Instruments | nstruments Ice type                     |       |
|-------------|------------------|-------------|-----------------------------------------|-------|
| 10 Feb 2024 | Saguenay 101     | D, G        | FSI. Consolidated ice fragments         | Calm  |
| 11 Feb 2024 | Saguenay 106     | D, B, S, G  | FSI Newly formed ice layer, homogeneous | Calm  |
| 20 Feb 2024 | Saint Fabien Bay | D           | Consolidated and fragmented             | Waves |
| 21 Feb 2024 | Baie du Ha! Ha!  | D, B, S, G  | Consolidated frazil melting             | Calm  |
| 23 Feb 2024 | Anse-à-Mercier   | D, B, S, G  | Consolidated frazil, ice in formation   | Waves |
| 26 Feb 2024 | Baie du Ha! Ha!  | D, B, S, G  | Consolidated frazil                     | Waves |
| 6 Mar 2024  | Lac St-Mathieu   | D, B, S, G  | Freshwater ice                          | Calm  |

#### 2 Instruments and methods

The BicWin 2024 field campaign is a series of multi-instrument acquisitions aimed at measuring simultaneously the mechanical properties of sea ice and the motion of ice induced by short-period (T 

**Figure 1.** a) Map showing the experiment sites in the Saguenay Fjord (stations 101 and 106), Lac Saint-Mathieu and the coastal area around the Bic National Park, Rimouski, Canada. b) A close-up view of Baie du Ha! Ha! and Anse-à-Mercier, which naturally collect sea ice pushed by dominant westerly winds. Bathymetric data are taken from the Canadian Hydrographic Service non-navigational (NONNA-10) product. The typical tidal range is about 3 m, with maximum values around 4 m.

**Figure 2.** a) Situation map of the 26 February around 19:00:00 UTC. Symbols present the different instruments deployed on the field. The scalebar corresponds to a horizontal distance of 10 m. b) Deployment timeline of the various instruments.

# 75 2.1 Geophones

The use of seismic waves to characterize the mechanical properties and the thickness of sea ice dates back to the late 1950s with pioneering works by Anderson (1958) and Hunkins (1960). However, such studies have remained rare because of the heavy logistics required in hostile, polar environments; there are therefore very few reports on in situ measurements of these parameters. Since the 2010s, however, new generations of instruments which significantly reduce the burden of seismic measurements have emerged: autonomous geophones can be installed on sea ice and record the seismic wavefield for several weeks (Moreau et al., 2020; Voermans et al., 2023), microphones can measure the air-coupled flexural wave to infer the ice thickness (Romeyn et al., 2021), and optical fibers can be deployed on floating ice for dense sampling of the wavefield (Nziengui-Bâ et al., 2022). Combined with appropriate signal processing and inversion strategies, the aforementioned authors have demonstrated the potential of these instruments to monitor the ice properties from passive seismic recordings. In itself, this achievement is of great value for future deployments of instruments able to transmit data remotely, thus providing direct measurements of the ice pack properties without the need of human intervention. During the 2024 BicWin campaign, our aim was twofold: i) to implement a protocol to rapidly characterize sea ice from active and/or passive measurements, for all ice conditions encountered including

**Figure 3.** a) Map showing the geographical location of the geophones for the four acquisitions on 11 February 2024: three acquisitions using linear arrays with 16 geophones (green circles), and one acquisition using a squared array with 12 geophones (blue circles). b) Relative positions of geophones within linear arrays with respect to the position of the active sources used on each side of the array. The six positions are numbered 101 to 106. Geophones are shown as circles while sources are shown as black squares.

thin, porous or soft ice; ii) provide reference seismic waveforms to validate those extracted with other instruments such as video recordings from unmanned aerial vehicles (see section 2.2), or wave buoys 2.3. In the next section, we describe the seismic array configurations used in our experiments. For our acquisitions, we used SmartSolo IGU-16HR 3C 5Hz smart sensors, with a sampling frequency of 1000 Hz)

#### 2.1.1 Data acquisitions

#### Linear arrays

Linear arrays are very convenient for extracting the dispersion relation of waves propagating in a medium. Dispersion curves relate the wavelength or wavenumber to the wave frequency, from which the phase and group velocities of the various propagating modes can be calculated at each frequency and inverted for estimating the material properties. Figure 3a shows three successive linear arrays arranged in a "U" shape. Each array contains 16 geophones separated by 3 m thus spanning a total length of 45 m. On each array side, we excite mechanical waves with local active sources, positioned at 5, 8 and 11 m away from the first and last geophones of the array, respectively. Each line is excited using 6 source positions, denoted 101 to 106

- (Figure 3b). Having multiple source positions is required for applying the methodology described in Serripierri et al. (2022). Three types of waves can propagate in a thin solid plate: shear waves, longitudinal waves, and out-of-plane waves denoted E (for east), N (for north) and Z (for vertical) in the context of a thin, horizontal ice sheet, with the north defined here as pointing in the direction of the linear array (see Figure 3b). In order to facilitate signal processing for extracting information out of these three different modes, sources are systematically produced at each location (101 to 106) following the procedure below, always in the same "ZEN" order.
  - 1. out-of-plane (Z) waves are generated by one person who jumps three times on the ice, each jump being a few seconds apart, in order to let waves propagate beyond the array.
  - 2. shear (E) waves are generated by hitting the ice with a hammer in the direction perpendicular to the linear array.
  - 3. longitudinal (N) waves are generated by hitting the ice with a hammer in the direction of the array.
- At each location 101 to 106, nine sources (three in each direction) are performed. Given the large variability of ice conditions in time and space, we designed this procedure to acquire sufficient seismic signals to characterize the ice in the shortest possible time. Each acquisition with a linear array typically takes 20 minutes for two people, including deployment, recording and removal of the geophones. We show in section 3.1 that we can extract from these recordings the averaged effective ice thickness, the Young's modulus, the Poisson's ratio and the density of the ice at each array position.

## Two-dimensional arrays

Two-dimensional (2D) arrays enable the use of more advanced array processing techniques, such as beamforming or tomographic inversions of the ice parameters (Rost and Thomas, 2002). In these experiments, we used quadrangular arrays to test strategies for tomographic inversions of the ice properties. To this end, we used active seismic sources, following an approach similar to the linear arrays source procedure. The sources were located on a circle having the same centre as the array. Figure 3a shows an example of a 2D array deployment on 11 February 2024. Contrary to the linear array, the 2D array has not been deployed every day, but only when the ice was strong enough to support active sources along the circle. In practice, we first performed the three linear array acquisitions and then proceeded with a 2D array when time allowed it and when the ice cover allowed it. We were able to deploy 2D arrays onFebruary 10, 11 and 26, as well as on March 6. On February 23 and 26, 2D diamond-shaped arrays were deployed, for passive seismic recordings. This geometry was chosen to optimize the sensitivity and the robustness of ice properties estimates in the presence of directional noise, such as the motion induced by a swell or residual wind waves.

#### 2.2 Unmanned aerial vehicles

Aerial observations of sea ice have been made in the Arctic and Antarctic regions since the 1970s. First used to perform reconnaissance flights to evaluate sea ice conditions and enable safer ship navigation, manned aircraft progressively became a powerful tool to study sea ice formation, evolution, interactions with surface waves (Martin and Kauffman, 1981), (Wadhams,

1975) and ice floe size distribution (Toyota et al., 2006). Aerial observations provide a wide range of fields of view, with a good temporal resolution, with field validation by the aircraft crew. However, operating a manned aircraft often requires large infrastructure and reasonable weather conditions and often implies prohibitive exploitation costs.

Satellites offer an operational platform for sea ice remote sensing and space-borne imagery is used to monitor sea ice conditions on large spatial scales. Ardhuin et al. (2015) for instance managed to estimate swell propagation in the marginal ice zone using Synthetic Aperture Radar (SAR) data from the Sentinel-1A satellite. However, they underlined that significant uncertainties remain in the directional spreading of the swell. Moreover, satellite observations of sea ice often lack temporal resolution, and concomitant field observations are often missing.

The recent development of lightweight unmanned aerial vehicles (UAVs) has enabled a high spatio-temporal resolution of aerial observations at low operational costs. Moreover, UAVs can be equipped with a large range of light-weight sensors such as RGB cameras, LiDAR, multispectral, thermal, or atmospheric sensors. Over the last decade, their use has spread in many fields of cryoscience to monitor snow cover, permafrost, glaciers, or even wild life evolution (Gaffey and Bhardwaj, 2020). Recently, UAVs were used to estimate sea ice floe size distribution (Zhang and Skjetne, 2015) in the Arctic. Dumas-Lefebvre and Dumont (2023b) also used an UAV to observe sea ice break-up by ship-generated waves and analyze the resulting FSD, while Sutherland and Gascard (2016) used an UAV equipped with a scanning LiDAR to measure surface waves propagation and attenuation in a marginal ice zone.

#### 2.2.1 Data acquisition

During this field campaign, we used two DJI Mavic 3 Pro to perform aerial observations. This UAV model can perform both photography and video acquisitions. The resolution varies from 12 to 20 MP. All videos were captured using the Hasselblad camera with a resolution of  $3840 \times 2160$  pixels at a frame rate of 30 Hz. The 4K-video format has been calibrated using the Camera Calibrator package developed on Matlab (Bouguet, 2022) for each drone. The calibration is based on the inference of the camera optical parameter from multiple images of a checker-board taken at various angles and distances. Looking at the deformations of the checker-board pattern at different locations on the sensor, we deduce both the camera focal length and the distortion induced by the camera lenses. To perform this calibration at a distance comparable to the field conditions, we replaced the checker-board by the tilling of a large building hall at Sorbonne University. We detected several rectangular tiles arrays on different pictures to get the camera characteristics. The measured focal length is  $f = 2696 \pm 15\,$  pix, and no significant optical radial distortion was found.

In the field, the flight information of the drone, including its location and orientation, and the camera state are monitored during each flight. Flight parameters are recorded either in the subtitle file associated with each video or in flight record files generated by the UAV controller. In practice, we use the drone GPS position  $(\phi_D, \lambda_D)$ , its vertical height  $H_D$  with respect to the water or sea ice elevation, the camera angle  $\alpha$  to the horizontal and the yaw angle  $\psi$  defined as the angle with respect to the geographic north. The vertical height  $H_D$  is measured thanks to a high precision barometric sensor, leading to a precision of about 0.1 m, which may go up to 0.3 m in windy conditions.

- During this campaign, we used UAV for i) documenting fieldwork activities (doc), ii) monitoring sea ice conditions using orthophotography (ortho), iii) characterizing wave propagation in sea ice (waves), and iv) observing wave-induced sea ice fracture (FRAC). During documentation and orthophotography operations, pictures or videos are collected without any particular restrictions on the camera pitching angle or drone height. On the other hand, movies of wave-ice interaction were captured during stationary flights above the ice edge at various heights ranging from 60 m to 200 m.
- The main purpose of each recordings defines the category of operation which are used to classify all images and videos recorded. The files associated to a given operation are gathered in a single folder named after the category of the performed operation (doc, ortho, waves, FRAC). A prefix and a suffix, corresponding respectively to the index over all types of operations and to the index over the operations of the same category, are added to the folder name.

Figure 4. Sample of the vertical acceleration  $a_z$  measured simultaneously by a wave-buoy (B2) and a smartphone (T1) on February 26, 2024. The comparison shows quantitative agreement.

## 2.3 Wave buoys

We used autonomous wave buoys to measure low frequency (<25 Hz) motions of ice, combining accelerometer, gyroscope, and magnetometer sensors. We deployed 6 high precision wave buoys (HPWB) developed by Peter Sutherland and his team (Veras Guimarães et al., 2018). We also tested the capacity of smartphones to be used as autonomous wave buoys (SWB for smartphone wave buoy): 6 to 30 smartphones were deployed for this purpose.

Ice motion recording with wave buoys pursued three goals. The first is to measure the wave propagation under ice as a function of time or space-dependent factors such as wind conditions, ice mechanical properties, water depth or distance to the ice edge. The second goal is to cross-validate ice motion recorded by the UAV. The third one is to test the reliability and accuracy of smartphone buoys in various wave conditions against HPWB.

We deployed buoys in arrays of various geometries to estimate wave dispersion and attenuation in the ice. Most of the data acquisition was performed using lines of wave buoys, oriented approximately in the main direction of propagation of the wave motion to focus on wave attenuation. We also deployed the instruments in other shapes, typically triangular shapes, to analyze the wave directionality. HPWB, SWB were also often deployed at the same location in several occasions, together with geophones and UAV recordings for multi-instrument cross-validation and integration.

#### 2.3.1 Data acquisition

Details about the sensors and the control of the HPWB wave buoys are described in (Veras Guimarães et al., 2018), while the details on the smartphone fleet are described in Zhang et al. (2025). We recall here only the main sensor characteristics. Each buoy measures simultaneously its acceleration in the three directions of space, angular velocity and orientation. The GPS signal eventually records the buoy position over time, with an horizontal accuracy of a few meters. The typical acceleration accuracy is 0.01 m s<sup>-2</sup>. HPWB are synchronized in time through the GPS chip, while the SWB has no automatic synchronization. SWB clocks are synchronized mechanically with a typical difference of 10 ms between different SWB. We synchronize the SWB

95 buoys with the SWB from the cross correlation function in time of a simultaneous recording of the ice motion at the same location. The smartphone temperature was maintained above 0°C thanks to a 3D-printed insulated box that keeps the phone temperature at about 15°C above the external temperature. We did not observe any alteration of the smartphone functionalities due to the cold.

Figure 4 shows a vertical acceleration signal  $a_z$  as a function of time recorded on February 26, 2024, comparing an HPWB(orange line, B2) and an SWB acquisition (blue line, T1). We observe typical wave packets, propagating under ice with a period  $T\sim 5$  s. We found a quantitative agreement between the two sensors, validating the measurement of waves with smartphone sensors. The typical wave vertical acceleration amplitude,  $a_z'\sim 0.1$  m s<sup>-2</sup> here is 10 times larger than the noise level.

**Figure 5.** Example of 3C data recorded after a source was made in the vertical direction on February 26, 2024 at position 102. Each trace is normalized by its maximum in the recording. Black, blue and red colours correspond to the vertical (Z), shear (E) and longitudinal (N) components, respectively.

#### 3 Results

This section provides some preliminary analyzes, showing promising results and demonstrating the potential of the synergy between all instruments.

## 3.1 Seismic data analysis

Figure 5 shows an example of seismic waveforms acquired on March 6 at the time of one vertical source at position 102, during a linear array acquisition. In the vertical channel, the flexural wave stands out, with high frequencies propagating faster than low frequencies. In the horizontal channels, however, the motions associated to shear and longitudinal waves are mixed. As the waveforms likely contain contributions from the three modes of propagation, it is therefore necessary to compute the dispersion curves to understand which modes are present. To compute the dispersion curves, we use the processing described in Serripierri et al. (2022). This processing takes advantage of the wavefield coherency through the array for the different sources. In short, it consists of applying a singular value decomposition (SVD) to the frequency spectra of all sources, before calculating

the wavenumber spectra with a Fourier transform on the space variable. This significantly improves the signal-to-noise ratio (SNR) in the dispersion curves (Minonzio et al., 2010).

The wavenumber vs frequency spectra shown in Figure 6 were obtained by applying this processing to all combinations of source directions and geophones channels. Since waves propagate only in one direction through the linear array, there is no negative wavenumbers. Knowing the direction of propagation, we unwrap the spectrum beyond the spatial aliasing limit. This technique was applied to produce Figure 6, which explains why the main branch is repeated vertically several times. Typically, the Fourier transform in the range 0 to 2 rad m<sup>-1</sup> is repeated between 2 and 4 rad m<sup>-1</sup>. We eventually identify the flexural wave branch on a larger bandwidth, which decreases the uncertainty on the parameter extraction.

The polarization of the flexural wave is in the sagittal plane (defined by the normal to the ice surface and the wavenumber), and its energy is mainly on the vertical direction (Moreau et al., 2020). Hence, we expect the energy of the flexural waves to appear mainly on the spectra of the vertical channel (Z), with residual signal on the longitudinal (N) channel. The polarization of the longitudinal wave is also in the sagittal plane, but it is dominated by the contribution of the N direction. Hence, one expects the longitudinal wave to appear mainly in spectra obtained from channel N, with residual signal on the spectra calculated from channel Z. Conversely, these two types of waves should only be generated by vertical or longitudinal sources. The shear horizontal (SH) wave, on the other hand, is polarized only along the direction E. Therefore, one expects to retrieve the spectral signature of shear waves from the channel E analysis. Similarly, the shear wave should only be generated by sources in the shear (E) direction.

Spectra in Figure 6 reveal that flexural waves are generated even with a E source, and it is visible on both N and Z channels. The longitudinal waves also appear in the spectrum of source E and channel N. We also note that the shear wave is recorded on channel E for all types of sources. To conclude, the method we use to generate different wave types from different source polarization, by either jumping onto the ice or hitting the ice with a hammer in a given direction was not fully selective. Nevertheless, the flexural, longitudinal and shear waves characteristics are contrasted enough to be separated during post-processing operations. The clear identification of the three wave types proves the efficiency of our protocol.

To recover the ice thickness, Young's modulus, Poisson's ratio and density from the dispersion curves, we use the forward model by Squire et al. (1996), based on the thin plate approximation, combined with Markov Chain Monte Carlo sampling of the parameter space, following Serripierri et al. (2022). However, instead of applying this methodology to dispersion curves obtained separately from the seismic waves generated in two opposite directions, we apply it to the average of the dispersion curves from the two propagation directions. The reason to do this is twofold. First, since there are no significant gradients of ice properties within the arrays, the averaged dispersion curves provide better SNR in the frequency-wavenumber spectra. Second, this averaging cancels undesired phase shifts that could potentially result from fluid layers with non-constant thicknesses Moreau et al. (2014), for example water layers trapped in the ice, or the snow layer. Inversion results are given in Table 2.

#### 3.2 AUV data analysis

The aerial pictures are geo-rectified using a backward projection method. To do so, we use the pinhole camera model (Dawson-Howe and Vernon, 1994) depicted in Figure 7a. In this model, the camera optic system is simplified to a camera pupil entrance

Figure 6. Wavenumber-frequency spectra of the acquisitions with the second linear array on February 26, 2024, for all combinations of source directions and geophone channels. Spectra were unwrapped by repeating the part of wavenumbers between 0 and 2 rad m<sup>-1</sup> from 2 to 4 rad m<sup>-1</sup>, in order to identify the flexural wave on a larger bandwidth.

and a sensor plane, separated by a distance f corresponding to the camera focal length. In this model, all light rays entering the camera are assumed to cross at the pupil entrance, located on the optical axis of the camera. We also assume that the surface to be reconstructed is horizontal. This approximation is reasonable for ice recording motions, as the distance from the camera to the surface is much larger than the out-of-plane motions of the ice. Using geometrical transformations, we associate to each pixel P on the camera sensor a point M on the ice surface. We attribute a pixel coordinate system  $(x_p, y_p)$  to the sensor plane with the origin taken as the image top-left corner. We note  $(x_0, y_0)$  the coordinate of the camera sensor centre. We also attribute a metric coordinate system (X, Y) to the ice plane, with the origin taken at the point corresponding to the image centre. The coordinates (X, Y) of a point M on the ice surface are related to the coordinates  $(x_p, y_p)$  of a pixel P on the camera sensor by

$$X = \frac{(x_p - x_0)H}{f\sin\alpha + (y_p - y_0)\cos\alpha}$$

$$Y = -\frac{(y_p - y_0)H}{\sin\alpha [f\sin\alpha + (y_p - y_0)\cos\alpha]},$$
(1)

**Table 2.** Sea ice thickness h, Young's modulus E, Poisson's ratio  $\nu$  and density  $\rho_{\rm ice}$  inverted from seismic acquisitions of all linear arrays, compared with ice thickness  $h_{\rm dh}$  measured from drill holes.

| Date        | Array # | $h_{\mathrm{dh}}$ (cm)                               | h (cm)        | E (GPa)        | ν               | $\rho_{\rm ice}~({\rm kg~m}^{-3})$ |
|-------------|---------|------------------------------------------------------|---------------|----------------|-----------------|------------------------------------|
| 10 Feb 2024 | 1       | 32 ±1                                                | $34 \pm 2.5$  | $5.3 \pm 0.8$  | $0.34 \pm 0.04$ | 887 ±140                           |
|             | 2       | $33.5 \pm 2.5$                                       | $33 \pm 2.5$  | $5.0\pm\!0.7$  | $0.36 \pm 0.04$ | $860 \pm 130$                      |
|             | 3       | $34.5 \pm 3.5$                                       | $37 \pm 2.5$  | $5.0\pm\!0.7$  | $0.36 \pm 0.04$ | $885 \pm 130$                      |
| 11 Feb 2024 | 1       | $11.5 \pm 0.5$                                       | $12\pm2$      | $2.4 \pm 0.5$  | $0.31 \pm 0.1$  | $895 \pm 165$                      |
|             | 2       | $11.5 \pm \hspace{-0.05cm} \pm \hspace{-0.05cm} 0.5$ | $12 \pm 2.5$  | $2.2 \pm 0.5$  | $0.31 \pm 0.1$  | $895 \pm 170$                      |
|             | 3       | $11.5 \pm \hspace{-0.05cm} \pm \hspace{-0.05cm} 0.5$ | $12 \pm 2$    | $2.2 \pm 0.5$  | $0.26 \pm 0.08$ | $875 \pm 165$                      |
| 26 Feb 2024 | 1       | $14\pm1$                                             | $13 \pm 3$    | $1.6 \pm 0.4$  | $0.38 \pm 0.10$ | $895 \pm 163$                      |
|             | 2       | $14 \pm \! 1$                                        | $16 \pm 4$    | $1.3 \pm 0.33$ | $0.38 \pm 0.10$ | $896 \pm 132$                      |
|             | 3       | $22\pm3$                                             | $25 \pm \! 6$ | $1.2 \pm 0.35$ | $0.38 \pm 0.11$ | $910 \pm 103$                      |
| 6 Mar 2024  | 1       | $43.5 \pm 2.5$                                       | 43 ±4         | $7.5 \pm 1.4$  | $0.39 \pm 0.05$ | 891 ±170                           |
|             | 2       | $43.5 \pm 2.5$                                       | $43 \pm \! 4$ | $6.8 \pm 1.3$  | $0.37 \pm 0.06$ | $871 \pm 157$                      |
|             | 3       | $43.5 \pm 2.5$                                       | $40 \pm 4$    | $7.2 \pm 1.3$  | $0.38 \pm 0.05$ | $898 \pm 160$                      |

where  $\alpha$  is the viewing angle with respect to the horizontal, and H the camera altitude with respect to the ice plane (see Figure 7). To calibrate the reconstruction technique, we deployed a star-shaped array of 13 red boxes easily detectable on the drone image in Baie du Ha! Ha! on March 2, 2024. We use decameters to set the edge length of each equilateral triangular mesh to 10 m. We performed a set of seven stationary flights with different camera angles  $\alpha$  and drone heights H. We then recorded a short video and extracted a picture of the star shape for each set of parameters  $(H,\alpha)$ . A picture captured at a height H=37.7 m and an angle  $\alpha=44.8^\circ$  is depicted in Figure 7b. We manually detected each smartphone box position on the images as indicated by the red circles of Figure 7b. The image distortion of the star-shaped array resulting from the inclined view is clearly visible.

Using the geometrical model, we associate the pixel position of the detected smartphones to their positions (X,Y) in the metric coordinate system, neglecting the curvature of the earth. Figure 7) shows the geo-rectified image together with the smartphone positions. The star formed by the smartphone array is now regular. We applied the projection algorithm to pictures taken from different positions, and we extracted the average edge length d of the star pattern for each parameter set  $(H,\alpha)$ . We found  $d=10.6\pm0.2$  m, compared with a target value of 10 m. The systematic error of about 5% may be attributed to the uncertainty on the UAV height H. Note that the uncertainty will also be a function of space, in particular for small  $\alpha$  values as we expect a larger uncertainty near the horizon line. From the geo-rectified image, the field of view can be geo-referenced by using the UAV GPS position at the record time and the drone orientation angles from the flight record. In practice, the uncertainty on the two angles introduce additional errors on the GPS location, of about 5 m for  $H \sim 100$  m. Whenever possible,

Figure 7. a) Schematic of the pinhole camera model. The camera is filming a plane from a height H, with an angle  $\alpha$  to the horizontal. Point M in the field plane corresponds to a point P on the camera sensor.(b) Image of smartphones deployment captured by the Hasselbald 4K camera objective. The UAV is at a height H=37.7 m above sea ice, the camera angle to the horizontal is  $\alpha=44.8^{\circ}$ . Red circles correspond to the star-shaped array of smartphones, manually detected on the image. The plane  $(x_p,y_p)$  corresponds to pixel coordinates on the camera sensor. (c) Backward projection of image (b) on the sea ice surface where the camera sensor centre is chosen as the centre of the (x,y) coordinate system.

the detection of several geo-referenced objects such as geophones, wave buoys or smartphones in the field of view provides a better localization.

The drone recording of wave motions can be used to extract the dispersion relation of surface waves in the presence of ice. Here we illustrate the procedure on one drone recording, performed on February 26, 2024 in Baie du Ha! Ha!. The video was taken with a vertical view at an altitude of 165.7 m for 3 minutes and 12 seconds, at a frame rate of 30 Hz. Figure 8a shows one image in physical scale. Open water appears on the left side, while continuous fast ice is visible on the right side. Small compacted ice fragments stand in between. After geo-rectification, we define a metric coordinate system with the origin at the bottom left corner. Thanks to the texture of sea ice fragments, we use a digital image correlation (DIC) algorithm (Pan et al., 2009) to efficiently compute the velocity field  $\mathbf{V}^*(x,y,t) = V_x^*(x,y,t) \hat{\mathbf{e}}_x + V_y^*(x,y,t) \hat{\mathbf{e}}_y$  seen from the drone camera point of view. DIC computes the local displacement between two images. To do so, it detects the motion of local patterns, using

Figure 8. a) Frame extracted from a drone flight performed over the ice edge in Baie du Ha! Ha! on February 26, 2024 with a vertical viewing angle at an altitude of 165.7 m. b) Demodulated wave field for a frequency f=0.212 Hz. c) Space-time spectrum  $\hat{V}_x(k,f)$  of the corrected velocity field where  $k=||\mathbf{k}||$ . The white dash line is the dispersion relation of open water waves with a water depth  $h_w=5.9$  m. Both axes and colormaps are in logarithmic scale. d) Linear dispersion relation f(k) extracted from the space-time spectrum. The open water dispersion relation is represented by the red dashed line. Both the x and y axes are logarithmic. The spectrum magnitude is represented by the colorbar. This figure was obtained using UAV Mesange (23waves012) on February 26, 2024 (48.34951°N, 68.81569°W) at  $t_0=20:35:59.647$  UTC.

square boxes of dimensions  $W \times W$  pixels. This correlation method is performed using the software PIVlab, developed in Matlab by Thielicke and Stamhuis (2014). To optimize the performance of the DIC, we compute the displacements between non-successive frames, separated by a number of frames  $\Delta t$ . In this test case, we used W = 32 pixels and  $\Delta t = 6$  frames.

The resulting velocity field is then post-processed to eliminate spurious velocity values. As the UAV is constantly adjusting its position to compensate the surrounding air flow, an additional field  $\mathbf{V}^{\text{drone}}(x,y,t)$  appears in the computed velocity field  $\mathbf{V}^{\text{drone}}(x,y,t)$ . Yet, this field differs from the surface wave field as the drone translational and orientational motions can be fitted by a 2D quadratic field  $V^{\text{drone}}_i(x,y) = a_0 + a_1x + a_2y + a_3xy + a_4x^2 + a_5y^2$ . We correct the total displacement field by fitting and subtracting for each time step a quadratic vector field  $\mathbf{V}^{\text{drone}}(x,y)$ . We attribute the remaining velocity field to the field associated to the wave propagation, noted  $\mathbf{V}(x,y,t) = V_x(x,y,t)\hat{\mathbf{e}}_x + V_y(x,y,t)\hat{\mathbf{e}}_y$ . This field contains a broad range of waves frequencies from 0.1 to 0.6 Hz. For each frequency, we extract a complex field  $\hat{\mathbf{V}}(x,y,t)$  filtered at one frequency. Figure 8b

**Figure 9.** Wave propagation under ice. a) Time series of vertical acceleration recorded with smartphone wave buoys (SWB) installed in a marginal ice zone in a line perpendicular to the ice edge on February 26, 2024. The colorbar indicates the distance from the first buoy. b) Phase diagram of the wave conditions measured with the smartphones for the three days (February 21, 23 and 26, 2024). Data are represented as a function of the local water depth h and the wave amplitude A. Colorbar codes the peak frequency of the associated wave spectrum.

shows the real part of the filtered wave field, for f = 0.212 Hz where one can see wave fronts parallel to the ice edge with a constant wavelength of about  $\lambda = 30$  m over the area where sea ice is made of slush and small fragments.

Using multidimensional Fourier transform, we compute the space-time spectrum of the x-component of the velocity field  $\hat{V}_x(k,f)$ , where  $k=|\mathbf{k}|$ . We then perform an average over the directions of the wave vector  $\mathbf{k}$ , to build the velocity spectrum  $\hat{V}_x(k,f)$  presented in Figure 8c. The spectrum exhibits a main branch corresponding to the dispersion relation of linear gravity waves in finite depth,  $\omega^2 = gk \tanh(h_w k)$ . The water depth was obtained using publicly available information about the local bathymetry (see Figure 1) and the tidal elevation with respect to the chart datum. The presence of sea ice does not alter significantly the dispersion relation. However, we observe wave attenuation, as shown on the demodulated field (Figure 8b). From the two dimensional spectrum, we extract the dispersion relation by computing the maximum of  $\hat{V}_x(k,f)$  for each frequency. The position of these maxima is refined by performing a second order polynomial fit with respect to the wavenumber k. We eventually obtain the dispersion relation as shown in Figure 8d.

## 3.2.1 Smartphone sensors data analysis

During the field campaign, we observed large spatial and temporal variations of the wave amplitude. As an illustration, Figure 9a shows the vertical acceleration  $a_z$  measured in 6 locations indicated in the situation map of February 26 (Figure 2a). The colorbar encodes the distance to the ice edge, whose position was inferred from a manual detection from the UAV video recordings. The wave amplitude decreased by a factor of about 2 over a 60 m distance. This spatial attenuation can be attributed to the wave damping by the ice sheet, which is significant for both continuous and fragmented ice.

Wave conditions experienced during the Bicwin 2024 campaign are eventually summarized on the phase diagram of Figure 9b for the three days during which significant wave heights were observed, namely February 21, 23 and 26. The water depth, obtained from the bathymetry and sea level data, ranges from 0.5 to 3 m while the wave amplitude varies from 2 mm to 20 cm. The peak wave frequency was mainly set by the wind conditions, with  $f_p=0.16\pm0.02$  Hz for February 21,  $f_p=0.32\pm0.05$  Hz for February 23 and  $f_p=0.21\pm0.03$  Hz for February 26.

#### 4 Code availability

The codes used to post-process the data are available at https://github.com/Turbotice/icewave.git This repository provides more than the code required to post-process and analyze the dataset introduced in the manuscript. In particular, we use it to process the data acquired during the BicWin campaign every year, as well as to analyze experimental data at the laboratory scale. The relevant module for processing specifically the Bicwin 2024 data corresponds to folders geophones, phones, and drones.

It should be noted that this repository is made available for open science purposes, but it is not made to be a tutorial for processing the data. A user guide is in development, but not yet available. In case of difficulties in extracting the scripts, please contact the authors with your inquiries

## 4.1 Geophones data processing scripts

For geophones data processing, the script plot\_smartsolo.py is for visualizing seismic data, waveform\_analysis\_from\_any\_source.py is to apply the specific processing introduced here (linear array acquisitions), and GPS\_coordinates\_from\_LOG\_files.py plots the positions of the geophones from their GPS coordinates.

# 330 4.2 AUV data processing scripts

Methods used to analyze aerial pictures and videos are described in different README.txt files. The different files obtained after a flight and their organization are described in the document README\_drone\_files.txt. One may find how to obtain the desired flight parameters by following this file. Pictures obtained from UAVs can be geo-rectified using the python package drone\_projection.py, the method is detailed in README\_georectification.txt. The script example\_georectification\_0306\_2024.py gives an example of how an image can be geo-rectified.

In order to process the Digital Image Correlation algorithm we used, one may be interested in the folder PIV\_processing which contains the two main scripts: automaticcycle\_banquise\_drone.m and Main\_data\_structuration.m. The use of these scripts is detailed in README\_PIV.txt while a few functions used to analyze the resulting velocity fields are detailed in README\_analysis.txt. The python script example\_analysis\_0226\_23\_waves\_012.py is given as an exemple for uploading DIC data and performing some methods presented in section 3.2

# 4.3 Smartphones data processing scripts

The main codes for processing the smartphone data are in the python submodule icewave.phone. The icewave.phone.load submodule contains the functions to download the raw data and agregate it in dictionnary. The functions to postprocess the data are located in the submodule icewave.phone.analyze, in particular to apply filtering, define a common timeline to all phones, compute Fourier transform, etc. Exemple notebook on how to process data from a specific day are also available in the git repository, for instance Compare\_Phone\_Buoys\_date.ipynb to perform mirror processing and comparison between smartphone and buoy time series, and Process\_cellphones\_date.ipynb to specifically analyze multiple. The keyword date in the notebook filename specifies the corresponding date in month day format of the campaign Bicwin24.

## 5 Data availability

The dataset Kuchly et al. (2025) is available at https://doi.org/10.57745/OUWL0Z. It is also archived at the St. Lawrence Global Observatory https://doi.org/10.26071/yvnd-2r11. Data are organized by date, and then by instrument. The data from AUV represent the main part in terms of volume, since we recorded our videos and photos in high resolution. The original dataset was about 2 Tb. Hence, a choice was made to include only the most relevant parts of our videos and photos, so that the dataset is kept at a reasonable size of about 100 Gb. All data are available upon request.

### 5.1 Geophones data structure

The "Geophones" folder contains i) the log files of the geophones for the date, and ii) sub-folders numbered 0001, 0002, 0003... that contain the recordings in minisced format. The number of a sub-folder corresponds to an acquisition configuration, as described in section. 2.2.1.

## 5.2 AUV data structure

Pictures and videos captured by UAVs are classified by device and then by flight category as detailed in section 2.2.1. Flight record files which gather the monitoring of several parameters during each flights are saved in the flightrecord folder. A few details of these files is given in the document README\_drone\_files.txt of the github repository presented above.

## 5.3 Smartphones data structure

The smartphone sensor data described in section 2.3.1 are classified by experiment name with the format CampaignNameYear\_RecordingName\_Year\_Date (exemple: Bic24\_S02\_2024\_0223). The recording name are in chronological order. The data of the smartphone sensors are contained in subfolders, with recording number, phone number, sensor type list and phone local date in the folder title. Each phone folder, contain a .csv per sensor (accelerometer, gyroscope, magnetometer, gps), as well as meta data provided by PhyPhox (Staacks et al., 2018). These data are agregated in one file, named \_phonedata.pkl.

## 6 Conclusion

We introduce a multi-instrument dataset designed to study mechanical wave propagation in fast ice, including its interaction with ocean swell and the propagation of seismic waves. This dataset was recorded at several locations of the St. Lawrence Estuary, Canada. Combined with original processing methodologies, it provides access to the complete process of sea ice breakup driven by gravitational waves, from the characterization of the ice's mechanical properties to the moment of breakup.

From seismic recordings, we extract the ice thickness, Young's modulus, Poisson's ratio, and density. The accuracy of the ice thickness estimations is particularly high, while the mechanical parameters are within the typical range for sea ice, although we observe substantial variability in Young's modulus depending on the meteorological conditions during ice formation — ranging from very soft, porous ice (E=1.2 GPa) to rigid ice (E=5.3 GPa). However, in cases of very soft ice where attenuation

dominates, or in partly consolidated ice where seismic waves cannot propagate, some acquisitions did not allow for parameter extraction. This is expected, since flexural motion cannot be transmitted through cracked ice.

From video recordings of aerial units, or arrays of inertial motion units (wave buoys and smartphones), we extract hydroelastic wave propagation excited by ocean waves. We developed a processing method to extract the vertical velocity field in continuous sea ice, assuming no horizontal displacement and a flat surface. We validate this approach by comparing the local displacement with IMUs measurements. It should be noted that, when the ice breaks, horizontal motion becomes possible between ice floes, which is current a limitation of our method. However, we intend to tackle this limitation by combining drone recordings with stereo particle image velocimetry. We hope that this will also provide a new way to measure longitudinal wave propagation, not only the flexural wave. We extract the main wave characteristics from the wave spectrum (amplitude, wavelength, attenuation). The method presents a sub-centimeter sensitivity on the vertical displacement, and allows access to the wave characteristics remotely. We demonstrate that the dispersion of ocean waves is recovered. The method can be applied to various types of sea ice – from very soft fast ice to rigid fast ice – and lake ice as wells as various sea conditions, from calm to agitated. This method could also extend seismic noise interferometry approaches to video recordings on sea ice, which would be a groundbreaking advance for sea ice monitoring.

The measurements of ice properties and wave characteristics are consistent with our direct observations in the field. Using the synergy of the multi-instrument data, and by renewing yearly the acquisitions, we aim to resolve the surface deformation prior and after break-up events, in both space and time. We will thereby track the propagation of fractures in the ice cover, in order to determine the rupture threshold associated with ocean waves forcing.

Video supplement.

Author contributions. All authors but VD participated to field measurements in the St. Lawrence Estuary. SK, BA, AE, PN, NM, DD, SP and LM organized, quality-controlled, processed and analyzed the datasets. LM processed the geophones data, and coordinated the writing of this paper. DD acted as Chief Scientist during the Amundsen expedition, and he organized and co-led field work activities with SP. All authors contributed to writing and reviewing the manuscript

Competing interests. The authors declare that they have no competing interests.

Acknowledgements. This work was funded by NSERC Discovery Grant *Physics of seasonal sea ice* (RGPIN-2019-06563) to DD and by the Mairie de Paris through Emergence(s) grant 2021-DAE-100 245973 (AE and SP), the Agence Nationale de la Recherche with the project

- Multiscale monitoring of sea ice parameters with passive seismology and deep learning, grant no. ANR-23-CE01-0020 to LM, SP and AE, and grant no. ANR-24-CE51-3840 to AE and SP. It contributes to the scientific program of the Quebec-Ocean Strategic Cluster funded by Fonds de recherche du Québec (FRQ) sector Nature and technology. Experiments in the Saguenay fjord were carried out as part of the Horizon glacé expedition onboard the CCGS Amundsen jointly funded by the PLAINE program of Réseau Québec maritime (RQM), the Transforming Climate Action program funded by the Canada First Research Excellence Funds (CFREF/Apogee) and by Amundsen Science.
- We are grateful to the Canadian Coastguard crew who facilitated the deployment of the ice canoe and all ice operation, and the staff of the Bic National Park for accommodation and support during the experiments. We also wish to thank participants to field work preparation and activities: Paul Nicot, Jérémy Baudry, Margaux Rougier, Vincent Denarié, Mathieu Plante, Baptiste Grison, Iga Vandenhove and Joan Sullivan.

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
