# Peer review of "An integrated multi-instrument methodology for studying marginal ice zone dynamics and wave-ice interactions"

_EGUsphere, 2025_

## Author Comment (AC1)

**Answer to reviewer 1**

In this manuscript, the authors discuss the use of several instruments to build a rich dataset that can be used to study waves in ice and wave-ice interaction in the MIZ. This is a timely and interesting study, and I am generally supportive of publication. I have a few comments that the authors may want to consider, see below.

We thank the reviewer for the useful comments, and we agree that the message of the manuscript can be unclear at parts, despite our efforts to make it as clear as possible. It is complicated to publish such a comprehensive dataset together with a few analyses in a same paper and keep the message clear. Our main intention is to introduce the dataset and field methodology, while also showing its potential with some preliminary results. We have prepared a detailed answer to the reviewer's comments, and modified our manuscript accordingly

- At present, the manuscript is a bit "strange" to read. At least it took me a couple of reads to really understand that this was only discussing methods. I think re-reading that this is relatively clear in the abstract, but could be made even clearer in the introduction. Similarly, the end of the paper feels very "abrupt", going from what feels a "methods" section directly to a conclusion and the end of the manuscript. I wonder if it may be worth to work a bit more on the presentation and structure of the manuscript, to group together the experiment description and data acquisition methodology on one hand, and the "preliminary analysis" / "proof of concept" that the data acquisition works well, on the other hand - i.e. creating a form of "introduction, methods, results" structure, which likely is closer to what the reader will expect.

We agree that the structure recommended by the reviewer for our manuscript could improve clarity, hence we have modified it to be more conventional:
Section 1 - Introduction
Section 2 - Instruments and methods
This section describes each instrument and the field methodology for our data acquisitions.
Section 3 - Results
This section provides some preliminary analyses, showing promising results and demonstrating the potential of the synergy between all instruments.
Section 4 - Code availability
This section was modified substantially to give more details about our processing scripts and which ones to use for each instrument type.
Section 5 - Data availability
This section was also modified to explain the structure of the dataset and more details about its content was provided.
Section 6 - Conclusions

- Similarly, I think that the authors should try to make it even clearer what the main values created by this manuscript are: in my opinion, but the authors may disagree of course: showing that the methodology works, but also and maybe as important providing all the data. The data on the https://entrepot.recherche.data.gouv.fr/dataset.xhtml?persistentI link are actually quite big - close to 100GB. This is not clear to the reader before visiting the URL, and should be presented and explained better: the present "Data Availability" section does not really convey this, neither I think does the text. It may also be worth spending more time discussing these data files, what is interesting in which one, which one contain what kind of conditions, etc.

We agree with the reviewer and have modified section 5 - Data availability - to give insight about the data structure. The new section is now as follows:

**5 Data Availability**

The dataset Kuchly et al. (2025) is available at https://doi.org/10.57745/OUWL0Z. It is also archived at the St. Lawrence Global Observatory https://doi.org/10.26071/yvnd-2r11. Data are organized by date, and then by instrument. The data from AUV represent the main part in terms of volume, since we recorded our videos and photos in high resolution. The original dataset was about 2 Tb. Hence, a choice was made to include only the most relevant parts of our videos and photos, so that the dataset is kept at a reasonable size of about 100 Gb.

**5.1 Geophones data structure**

The "Geophones" folder contains i) the log files of the geophones for the date, and ii) sub-folders numbered 0001, 0002, 0003... that contain the recordings in miniseed format. The number of a sub-folder corresponds to an acquisition configuration, as described in section.

**5.2 AUV data structure**

Pictures and videos captured by UAVs are classified by device and then by flight category as detailed in section 2.2.1. Flight record files which gather the monitoring of several parameters during each flights are saved in the flightrecord folder. Some details of these files are given in the document README_drone_files.txt of the github repository presented above.

**5.3 Smartphones data structure**

The smartphone sensor data described in section 2.3.1 are classified by experiment name with the format CampaignNameYear_RecordingName_Year_Date (exemple : Bic24_S02_2024_0223). The recording name are in chronological order. The data of the smartphone sensors are contained in subfolders, with recording number, phone number, sensor type list and phone local date in the folder title. Each phone folder, contain a .csv per sensor (accelerometer, gyroscope, magnetometer, gps), as well as meta data provided by PhyPhox (Staacks et al., 2018). These data are agregated in one file, named _phonedata.pkl.

However, we would like to refrain from providing the level of description suggested by the reviewer, as this is not our aim—particularly since the aspects that are of interest to us may differ from those that would be of interest to others. Regarding acquisition conditions, an overview of the ice type and sea state is provided in table 1, and we cannot provide more detailed conditions.

- I also miss a more "personal" / "analysis-focused" discussion of the data. It is already nice to read that you were able to perform these measurements and that this is doable; however to be truly useful to me, I would need to know more: i) the technical details (hardware and software used, see below) so that I can really reproduce all or part of the measurements in my user case. ii) Maybe even more important, I miss an honest, frank discussion about the different methods: what works well or not so well in the experience of the authors? What are the "tricks and tips" the authors have discovered from their field experiments (I think this is quite key in a paper that is focused on method rather than scientific results per se). What are the limitations and caveats of the methods presented here? I think this is a key area in which the authors can provide added value for the reader and the field, and that this is not fully covered now.

Regarding the first point (technical details), we have focused on detailing section 4 - Code availability -, which has been modified as follows:

**4 Code availability**

The codes used to process the data are available at `https://github.com/Turbotice/icewave.git`. The repository is still at a development stage, and thus also contains parts that are not only related to the dataset introduced in the manuscript. We acquire new data every year, as well experimental data at the laboratory scale. The parts relevant for the processing of the present dataset are in folders geophones, phones, and drones. Our scripts are homemade in Python and Matlab languages.

It should be noted that this repository is made available for open science purposes, but it is not made to be a tutorial for processing the data. A user guide is in development, but not yet available. In case of difficulties in extracting the scripts, please contact the authors with your inquiries

**4.1 Geophones data processing scripts**

For geophones data processing, the script plot_smartsolo.py is for visualizing seismic data, waveform_analysis_from_any_source.py is to apply the specific processing to these data, and GPS_coordinates_from_LOG_files.py plots the positions of the geophones from their GPS coordinates.

**4.2 AUV data processing scripts**

Methods used to analyse aerial pictures and videos are described in different README.txt files. The different files obtained after a flight and their organization are described in the document README_drone_files.txt. One may find how to obtain the desired flight parameters by following this file. Pictures obtained from UAVs can be geo-rectified using the python package drone_projection.py, the method is detailed in README_georectification.txt. The script example_georectification_0306_2024.py gives an example of how an image can be geo-rectified. In order to process the Digital Image Correlation algorithm we used, one may be interested in the folder PIV_processing which contains the two main scripts : automaticcycle_banquise_drone.m and Main_data_structuration.m. The use of these scripts is detailed in README_PIV.txt while a few functions used to analyse the resulting velocity fields are detailed in README_analysis.txt. The python script example_analysis_0226_23_waves_012.py is given as an exemple for uploading DIC data and performing some methods presented in section 3.2

**4.3 Smartphones data processing scripts**

The main codes for processing the smartphone data are in the python submodule icewave.phone. The icewave.phone.load submodule contains the functions to download the raw data and agregate it in dictionnary. The functions to postprocess the data are located in the submodule icewave.phone.analyze, in particular to apply filtering, define a common timeline to all phones, compute Fourier transform, etc. Exemple notebook on how to process data from a specific day are also availaible in the git repository, for instance Compare_Phone_Buoys_date.ipynb to perform mirror processing and comparison between smartphone and buoy time series, and Process_cellphones_date.ipynb to specifically analyze multiple. The keyword date in the notebook filename specifies the corresponding date in month day format of the campaign Bicwin24.

Regarding the second point (discussion about the different methods and their limitations), we are not sure that there is much more to say than what is already explained in the manuscript. The limitations of seismic methods are already covered in the literature, and we have recalled, in section 3 as well as in the conclusions, that recordings were

impacted when the ice was fractured because the flexural wave cannot propagate in fragmented ice, preventing the extraction of sea ice parameters in some cases. However, we have also added, in the conclusion, a current limitation to our video processing method:

It should be noted that, when the ice breaks, horizontal motion becomes possible between ice floes, which is current a limitation of our method. However, we intend to tackle this limitation by combining drone recordings with stereo particle image velocimetry. We hope that this will also provide a new way to measure longitudinal wave propagation, not only the flexural wave.

- A minor language / taste point: maybe be careful of too strong formulations; for example "Wave-driven fragmentation is the key mechanism shaping the Marginal Ice Zone (MIZ).": I agree that it is a key mechanism, but is it really *the* mechanism, in all conditions? Sometimes, the MIZ is broken from before, and then currents and winds play a major role for example.

We thank the reviewer for spotting this inaccuracy, we have changed the text.

- Regarding geophones: as you point out, there have been a lot of developments recently, in particular regarding cost. A possible reference point on this is the recent work of Voermans et al. https://doi.org/10.1017/jog.2023.63 in which it was possible to deploy geophones for a cost of O(500USD) per geophone, and to infer back information about the sea ice properties passively. Can this be relevant for your future expeditions too, or are you limited to using active sources anyways due to the lack of other naturally occurring signal in your area of interest? How does your solution compare to what is described there?

This is an interesting question. Although the manuscript only presents preliminary results from active sources, as mentioned in the data acquisition section, we also performed passive recordings to recover the ice properties from ice swell or other types of naturally existing sources, such as icequakes for example. Although active sources provide a quick and efficient way of characterizing the ice, one should not rely too much on such methods in the open Arctic, where logistics are a limiting factor and should therefore be kept to a minimum. We will soon introduce new methodologies based on our passive data acquisitions, that will expand passive approaches already introduced by Moreau et al. (2020a,2020b, 2023), and Serripierri et al. (2022). Although we would like to, we cannot cover all possibilities offered by such a dataset in this manuscript only, because there is so much more to say for all instrument types. Instead, this manuscript will serve as a reference to introduce new results without having to describe the data in upcoming new manuscripts. We have included the reference Voermans et al. (2023) in our manuscript, and thank the reviewer for the suggestion.

To answer the specific question about costs, the Smartsolos geophones are a bit more expensive than 500USD, but the difference in price does not justify the burden of having to go through the whole development of a new instrument in the context of our study. However, we agree that the development of specific instruments, including seismic sensors specifically for sea ice, should play an important role in the future. This is a reason why we also focus on optical methods with AUV, and also on Distributed Acoustic Sensing (which was included in our 2025 field campaign).

- Some of the instruments are described in quite a bit of details (for example, the UAV data acquisition setup - there is enough information that I could likely find the hardware and software I should use if I wanted to reproduce similar measurements), but for some others, the information is quite limited (for example, the geophones and signal source design / models used are not clear to me?), as discussed above. Consider adding an Appendix with a summary of all the hardware and software used, for example in a series of tables, to make the technical aspects easier to reproduce and investigate in more

The revised version of section 4 – Code availability – now includes guidelines about which scripts to use to process each type of data. We have also added, in section 2.1 – Geophones, the type of geophones used in our field campaign:

For our acquisitions, we used SmartSolo IGU-16HR 3C 5Hz smart sensors, with a sampling frequency of 1000 Hz)

We are not sure of what is meant by "signal source design", but we have not used any specific seismic sources other than those described in section 2.1.1.

Once again, we would like to thank the reviewer for their useful comments, and we hope that the changes made to

the manuscript, together with our answers to the comments, have clarified things and improved the manuscript.

---

## Author Comment (AC2)

**Answer to reviewer 2**

*An impressive and complete methodology and dataset! Looking forward to where this goes next and the new avenues it opens for more data on wave-induced sea ice fracture!*

We thank the referee for their reading and comments as well as their enthusiasm for our methodology and dataset. Please find below our answers to the questions raised by the referee.

*Can you comment on whether this dispersion relation was expected? i.e. that the presence of sea ice did not change the dispersion relation relative to that which we expect from open water? If so, why? Was it due somehow to this specific sea ice type? I ask because I know that there is quite a lot of literature regarding developing dispersion relations which do consider the presence of sea ice (i.e. with the implication that the presence of sea ice can modify the dispersion relation).*

During the Bicwin24 campaign, we observed and characterized both ice floes, and continuous ice of thickness ranging from 10 to 25cm (except for the 11/02 in Saguenay fjord). Figure 7a is representative of the sea ice conditions near the ice edge we observed during the Bicwin24 campaign. The ice edge was composed of small ice fragments ranging from a few centimetres to less than 5 meters large, and the thickness $h \sim 16$ cm. Most of the waves energy was carried by waves with a typical wavelength ranging between 20 and 30 meters. In this configuration, all fragments have sub-wavelength dimensions, and therefore, do not modify significantly the dispersion relation. However, these fragments generate dissipation of the wave energy.

Further from the ice edge, the ice is continous and may affect the dispersion relation. In the presence of a floating elastic plate of thickness $h$, density $\rho$ and flexural modulus $D$ above a water column of height $h_w$, and neglecting dissipative effects, the most general dispersion relation of hydro-elastic waves that can be found in the literature [1, 2] writes :

$$\omega^2 = \frac{\left(gk + \frac{D}{\rho_w}k^5\right)\tanh(kh_w)}{1 + kh\frac{\rho}{\rho_w}\tanh(kh_w)} \tag{1}$$

We can evaluate the different terms of the above dispersion relation in our field conditions. The typical wavelength (20 to 30m) is much greater than the ice thickness, leading to $kh \sim 4.5 \times 10^{-2}$. The term $hk\frac{\rho}{\rho_w}\tanh(kh_w)$ associated to sea ice buoyancy can therefore be neglected. We can also compare the gravity and elastic terms using the gravito-elastic length $l_D = \left(\frac{D}{\rho_w g}\right)^{1/4}$. Considering a Young modulus $Y = 1.5$ GPa and Poisson coefficient of $\nu = 0.38$ as measured on February 26 (table 2 of the manuscript), the gravito-elastic length equals $l_D \sim 4.7$ m. This typical length is much smaller than the wavelength, meaning that the observed waves are in a purely gravitational regime as represented in Figure 7c and 7d. The measured water depth $h_w = 5.9$ m influences the wave propagation, and the dispersion relation indeed shows both shallow water and deep water regimes.

**SPECIFIC COMMENTS**

- *"Therefore, achieving accurate and multi-scale estimates of ice parameters will set the new standards of sea ice monitoring." Consider rewording, this is a bit hard to follow.*

  We have modified the sentence as follows

  Therefore, obtaining accurate, multi-scale estimates of ice parameters will establish a new benchmark for sea ice monitoring.

- *Regarding figure five (we assume the reviewer means Figure 6): "The polarization of the flexural wave is in the sagittal plane." This is the out-of-plane wave, correct? Is it necessary to have two different naming conventions? If so, the first time "flexural wave" is introduced, maybe put "out-of-plane wave" in brackets next to it.*

  We agree that sagittal plane can be a bit misleading, so we have modified the text to clarify that:

  The polarization of the flexural wave lies in the sagittal plane (defined by the normal to the ice surface and the wavenumber), with its energy mainly oriented in the vertical direction.

  However, it should be noted that the longitudinal wave is also polarized in the same plane. Therefore, it makes little difference whether we refer to in-plane or out-of-plane motion, since both waves contribute to these motions—albeit in different proportions. We hope this clarifies things a bit.

- *It would be helpful to guide the reader a bit more if you included panel labels which are referred to in the text (Fig. 5a–i).*

  We assume the reviewer means figure 6. Yes, thank you for the suggestion. We have added panel labels in Figure 6, and referred to each of them in the analysis of Figure 6.

- *"where on can see" should be "where one can see".*

  Thank you for spotting this typo. It has been corrected.

- *Figure 9 is missing "a), b)" labels on the panels despite being referred to in the caption.*

  Thanks again! This has been corrected.
* * *
[1] A. G. Greenhill, Wave Motion in Hydrodynamics, American Journal of Mathematics **9**, 62 (1886).

[2] H. H. Shen, enWave-in-ice: theoretical bases and field observations, Philosophical Transactions of the Royal Society A: Mathematical, Physical and Engineering Sciences **380**, 20210254 (2022).